# The Chaperone System in Breast Cancer: Roles and Therapeutic Prospects of the Molecular Chaperones Hsp27, Hsp60, Hsp70, and Hsp90

**DOI:** 10.3390/ijms23147792

**Published:** 2022-07-14

**Authors:** Giusi Alberti, Giuseppe Vergilio, Letizia Paladino, Rosario Barone, Francesco Cappello, Everly Conway de Macario, Alberto J. L. Macario, Fabio Bucchieri, Francesca Rappa

**Affiliations:** 1Department of Biomedicine, Neurosciences and Advanced Diagnostics (BiND), University of Palermo, 90127 Palermo, Italy; giusi.alberti@unipa.it (G.A.); giuseppe.vergilio@unipa.it (G.V.); rosario.barone@unipa.it (R.B.); francesco.cappello@unipa.it (F.C.); fabio.bucchieri@unipa.it (F.B.); francesca.rappa@unipa.it (F.R.); 2Euro-Mediterranean Institute of Science and Technology (IEMEST), 90139 Palermo, Italy; ajlmacario@som.umaryland.edu; 3Department of Microbiology and Immunology, School of Medicine, University of Maryland at Baltimore-Institute of Marine and Environmental Technology (IMET), Baltimore, MD 21202, USA; econwaydemacario@som.umaryland.edu

**Keywords:** breast cancer, chaperone system, molecular chaperones, Hsp27, Hsp60, Hsp70, Hsp90, negative chaperonotherapy, Hsp inhibitors, immunotherapy

## Abstract

Breast cancer (BC) is a major public health problem, with key pieces of information needed for developing preventive and curative measures still missing. For example, the participation of the chaperone system (CS) in carcinogenesis and anti-cancer responses is poorly understood, although it can be predicted to be a crucial factor in these mechanisms. The chief components of the CS are the molecular chaperones, and here we discuss four of them, Hsp27, Hsp60, Hsp70, and Hsp90, focusing on their pro-carcinogenic roles in BC and potential for developing anti-BC therapies. These chaperones can be targets of negative chaperonotherapy, namely the elimination/blocking/inhibition of the chaperone(s) functioning in favor of BC, using, for instance, Hsp inhibitors. The chaperones can also be employed in immunotherapy against BC as adjuvants, together with BC antigens. Extracellular vesicles (EVs) in BC diagnosis and management are also briefly discussed, considering their potential as easily accessible carriers of biomarkers and as shippers of anti-cancer agents amenable to manipulation and controlled delivery. The data surveyed from many laboratories reveal that, to enhance the understanding of the role of the CS in BS pathogenesis, one must consider the CS as a physiological system, encompassing diverse members throughout the body and interacting with the ubiquitin–proteasome system, the chaperone-mediated autophagy machinery, and the immune system (IS). An integrated view of the CS, including its functional partners and considering its highly dynamic nature with EVs transporting CS components to reach all the cell compartments in which they are needed, opens as yet unexplored pathways leading to carcinogenesis that are amenable to interference by anti-cancer treatments centered on CS components, such as the molecular chaperones.

## 1. Introduction

Laboratory and clinical research on breast cancer (BC) has been intensive for many years because of the high frequency and malignancy of the diverse forms of this tumor. However, there are still various unresolved issues that need clarification for developing accurate and fast diagnostic tests, reliable disease-monitoring strategies, and efficacious treatments. For example, the role of the chaperone system (CS) in tumor initiation, progression, and resistance to treatment, as well as in anti-tumor mechanisms, is still poorly understood. Despite the many reports on the levels of molecular chaperones, i.e., the chief components of the CS, in tumor tissue and a variety of samples from patients, and despite experiments to determine their role in the tumor’s biology, there are still key mechanisms in which chaperones and the other components of the CS most likely participate that have not been elucidated [1]. We have undertaken research on the role of the CS in various tumors starting from histological studies to quantify and map diverse chaperones in the tumor tissue and continuing with the detection of chaperones in EVs released by the tumors [2,3]. It has become increasingly clear that chaperones do not act alone, but rather in conjunction with the other components of the CS. The latter is composed of molecular chaperones, chaperone co-factors, co-chaperones, and chaperone interactors and receptors [1].

The chaperones typically form teams that interact to build functional networks for the maintenance of protein homeostasis in the entire body, as well as in tumor tissue. Furthermore, in performing its canonical functions pertaining to protein homeostasis, the CS collaborates with the Ubiquitin–Proteasome System (UPS) and the chaperone-mediated autophagy (CMA) machinery. The CS also displays non-canonical functions unrelated to protein quality control but pertaining to other mechanisms. For instance, the CS interacts with the immune system (IS) in performing some of its non-canonical functions in cancer and inflammatory and autoimmune conditions [1]. It follows that the study of one or a few molecular chaperones and the estimation of their value as biomarkers in BC do not cover the entire range of their capabilities and the mechanisms in which they participate. The interpretation of results is difficult, if not impossible, because of incomplete information. Therefore, developing treatments based on those studies circumscribed to one or a few chaperones and ignoring the other components of the CS and its main interactors, i.e., UPS, CMA, and IS, may be disappointing, as already demonstrated by a number of investigators. Consequently, we have implemented projects taking into account the complexity of the CS and its functional connections in the body, even if the research must proceed one step at a time, because it is unfeasible to study all the components of the CS simultaneously. The key point to bear in mind is that analyzing a single chaperone will produce results that must be interpreted considering the entire CS and its currently known functions as a physiological system. In this review, we surveyed works dealing with four molecular chaperones, Hsp27, Hsp60, Hsp70, and Hsp90, as well as their roles and possible uses in therapeutics in breast cancer. The main aim was to produce a repository of information that would be useful for future research and informative to practitioners of medicine managing BC patients. It is hoped that this review becomes a launching platform for future studies on the participation of the CS in BC, reaching beyond the four chaperones examined here to encompass other members of the CS and its associated systems, such as the IS and UPS, and the CMA machinery.

BC is a heterogeneous disease with several subtypes of different cellular compositions, molecular alterations, and clinical manifestations [4]. Many factors are generally considered in prognostication: histological grade, tumor type and size, lymph node metastases, estrogen receptor (ER), progesterone receptor (PR), and human epidermal growth factor receptor 2 (HER2), but these are insufficient indicators for the proper clinical management of patients [5,6]. For example, the identification and characterization of reliable biomarkers, such as the components of the CS, must be explored.

The chief components of the CS are the molecular chaperones, which can be classified in groups according to their molecular weights, encompassing the following ranges in kDa: ≤34, 35–54, 55–64, 65–80, 81–99, 100–199, and ≥200; within these groups are families of phylogenetically related molecules termed ‘heat shock proteins’ (Hsps), such as the Small Hsp (sHsp; those with the alpha-crystallin motif), Hsp40/DnaJ, Hsp70/DnaK, and Hsp90 families, and others, e.g., the CCT (Chaperonin-containing TCP1) family [7]. A proposal for the nomenclature of some components of the CS is available [8].

In this review, we will discuss one representative of the sHsps (Hsp27), Hsp60, and examples involving members of the Hsp70 and Hsp90 families. Although molecular chaperones are typically cytoprotective, they may also become etiopathogenic factors if abnormal in structure-function, location, or concentration, causing diseases, i.e., the chaperonopathies [9]. These include several types of tumors such as breast, colon, lung, and other cancers [2,3,7]. In cancer cells, the chaperonome is extensively remodeled and participates in their altered metabolism, protects them from stressors, and helps them to resist or elude the IS [10,11]. The expression of Hsps has been associated with tumor-cell proliferation and differentiation, as well as with resistance to apoptosis and poor prognosis [12]. Numerous observations demonstrate that Hsps are augmented in BC tissue, with the degree of increase paralleling the degree of malignancy [13]. Hsp27, Hsp60, Hsp70, and Hsp90 are promoters of tumorigenesis in BC and are implicated in pathogenic mechanisms underpinning invasiveness, epithelial-mesenchymal transition (EMT), metastasization, and resistance to therapy [13]. Therefore, any treatment that can target and block/inhibit these Hsps (i.e., negative chaperonotherapy) in BC cells has potential clinical applications. In this review, various inhibitors (natural and synthetic) of Hsps are discussed. EVs carrying Hsps, anti-Hsp antibodies, and Hsp-based anticancer vaccines are also examined to provide a reasonably complete matrix of information on the pathogenic roles and potential in anti-BC therapeutics of Hsp27, Hsp60, Hsp70, and Hsp90.

## 2. The Biology and Heterogeneity of BC

BC is the most common type and leading cause of cancer death among women [14]. The origin of BC is complex, because several genetic and epigenetic changes are involved, which are the cause of the known histopathological–clinical heterogeneity of this cancer [15]. Intertumoral heterogeneity has led to a classification based on expression profiles of key molecular markers, such as ER, PR, and HER2 [16]. Hence, BC is classified into four subtypes considering their receptor statuses: luminal A (ER-positive, PR-positive, and HER2-negative); luminal B (ER-positive and/or PR-positive, HER2-positive); HER2 overexpressing (ER-negative, PR-negative, and HER2-positive); and triple negative (ER-negative, PR-negative, HER2-negative) (Figure 1).

Data from the American Cancer Society and the American College of Surgeons collected since 2010 show that 14.5% of 298,937 invasive BC cases were HER2-positive [17]. HER2-positivity is generally considered to predict aggressive phenotypes and poor prognosis, because HER2-amplified tumors are larger and have a higher number of positive nodes and a higher lympho-vascular invasion rate than HER2-negative tumors (Figure 1) [18,19]. BC arises from the epithelial lining of the ducts or lobules. Therefore, the typing of invasive BC and its histological variants divides this tumor into the categories of ductal, lobular, and not otherwise specified (NOS). Significant heterogeneity in BC further subcategorizes it into in situ and invasive carcinomas. In general, breast lesions are sorted into ductal carcinomas in situ (DCIS) and invasive ductal carcinomas (IDC) (Figure 1). DCIS is a noninvasive intraductal proliferation of epithelial cells that is confined to the ducts and lobules, whereas IDC shows abnormal malignant proliferation of neoplastic cells in breast tissue, following migration into the stroma through the duct wall [20]. Furthermore, according to the site where the tumor originates, BCs can be divided into ductal and lobular [20]. DCIS is considered the precursor to IDC and presents several distinct and identifiable morphologies classified into five subtypes, namely, comedo, solid, cribriform, papillary, and micropapillary. DCIS presents different histological types with some additional morphological features, such as the presence of multinucleated giant cells, nuclear atypia, intraluminal necrosis, mitotic activity, and calcification; depending on the degree of these features it is possible to distinguish IDCS of low, intermediate, and high grade [21]. DCs are breast tumors with malignant ductal proliferation together with invasion of the stroma in the presence or absence of DCIS [20] and are classified into many histological subtypes according to a variety of criteria, including the cell type (apocrine carcinoma); amount, type, and location of secretion (mucinous carcinoma); histological features (papillary, tubular, and micropapillary carcinoma); and immunohistochemical profile (neuroendocrine carcinoma) [22]. The most common type of IDCs is the invasive ductal carcinoma not otherwise specified (IDC-NOS) or of no special type (IDC-NST), which is a diagnosis of exclusion and comprises adenocarcinomas that do not show sufficient characteristics to warrant their classification into one of the special types [23]. Invasive lobular carcinoma (ILC) constitutes 5–15% of invasive BC and usually occurs in older women affected by conventional IDC [24]. Lobular carcinoma includes lobular carcinoma in situ (LCIS) and invasive lobular carcinoma (ILC) (Figure 1). ILC accounts for 5% to 15% of invasive BCs, usually occurs in the older age group of women, and consists of small, round, relatively uniform cells with a characteristic growth pattern with infiltration into the stroma [25]. Hallmark molecular features of ILC include, above all, inactivation of E-cadherin by mutation, loss of heterozygosity, or methylation, but not β-catenin positivity for both the ER and PR or HER-2 negativity [26,27]. The mutational inactivation of E-cadherin is an oncogenic driver in ILC [28,29]. The World Health Organization’s (WHO) classification of tumors of the breast (5th edition, 2019) recognizes ILC as the most common special type of BC [30]. These heterogeneous pathological features and the diffuse growth pattern of ILC make diagnosis particularly challenging [31]. ILC tumors show diverse histological variants, including classic, solid, alveolar, mixed, tubulo-lobular, and pleomorphic lobular carcinoma [24]. On the contrary, LCIS consists of neoplastic cells that fill and expand over 50% of the acini with round to oval shapes and inconspicuous cytoplasm [32]. Similar to ILC, LCIS is highly ER-positive, while PR expression is lower, and only 11% of tumors have HER2 amplification. Currently, histological features are the basis for the classification of LCIS into three main subclassifications: classical (CLCIS), florid (FLCIS), and pleomorphic (PLCIS), and these types can coexist in the same patient [33].

Overall, BCs are heterogeneous, both histologically and molecularly, so traditional BC treatments depend on tumor characteristics, such as the clinical stage, histopathological features, and biomarker profile. However, the subtyping of BC is still challenging and controversial. Therefore, accurate stratification is required to plan adequate treatment, since each group or subgroup has a distinctive prognosis and requires individualized systemic therapy. We hope that by quantifying and mapping the components of the CS, such as the Hsps discussed in this review, in BC tissue will add information of value to differentiate tumor types reliably. Furthermore, analysis of the Hsps in the EVs released by the BC mass is likely to provide valuable sequential information over time because minimally invasive sampling can be performed repeatedly.

## 3. Quantitative Levels and Functions of Hsp27, Hsp60, Hsp70, and Hsp90 in BC

### 3.1. Hsp27

Hsp27 (HSPB1) is a CS member belonging to the sHsp family (≤34 kDa) and is a redox-sensitive molecular chaperone. The primary structure of Hsp27 is highly homologous to other members of the sHsp family that bear the conserved α-crystallin domain (ACD) and variable-length N- and C-terminal regions (NTRs and CTRs, respectively) [34]. Hsp27 is an ATP-dependent chaperone that folds substrate proteins and participates in shuttling destabilized or misfolded proteins toward proteolytic degradations [35]. Hsp27 is cytoprotective; it blocks apoptosis through interaction with various factors active in the apoptotic pathway [36]. The inhibition of apoptosis is required in the heat shock response (HSR) for cells to survive transient stress-induced apoptosis accompanied by the Hsp-mediated refolding of damaged proteins [37] (Figure 2).

Hsp27 is expressed in all human tissues and shares the properties of being phosphorylated and oligomerized with other members of the sHsp family [38]. Unphosphorylated Hsp27 forms multimers that can reach 800 kDa, representing the binding-competent state with an affinity for client proteins, and phosphorylation causes conformational changes leading to a decrease in oligomer size, followed by dissociation into monomers with the loss of chaperone activity [39]. The phosphorylation status is closely linked to diseases such as cancer and diabetes [39,40]. In cancer, phosphorylation changes the affinity of Hsp27 for its client oncoproteins, leading to the activation of anti-apoptotic and pro-survival signaling pathways [41,42] (Figure 2). The overexpression of Hsp27 in cancer cells is implicated in tumor growth, metastasization, and the induction of chemoresistance, through the stabilization of various oncogenic genes and proteins involved in tumor progression [11]. The association of Hsp27 with BC, especially the oncogenic properties and drug responsiveness of the HER2^+^ subtype, has been shown [43,44]. In BC patients with HER2-overexpression, HSF1-mediated Hsp27 upregulation and Ser15 phosphorylation lead to increase in HER2 nuclear function, reducing susceptibility to trastuzumab (TZMB) [45]. In HER-2/neu positive and negative BCs, highly phosphorylated Hsp27 was found on Ser78 and driving or facilitating in vivo tumor-cell invasion and metastasis [46]. Moreover, biopsies from patients with lymph node BC metastasis showed significantly higher levels of phosphorylated Hsp27 molecules in the nucleus and cytoplasm compared with a group without lymph node metastasis [40]. Hsp27 levels were significantly higher in patients with BC than in the control group [47,48]. Studies of in vivo metastasization revealed that Hsp27 is overexpressed in the aggressive forms of human BC, with a 2.57-fold increase over the median in cancers that are positive for lymph node invasion and a 2.95-fold increase over the median in the metastatic ones, pointing to Hsp27 as a potential therapeutic target for negative chaperonotherapy in BC with bone metastases [49]. Compared with hormone receptor or HER2-positive BC, triple-negative BC has a higher percentage of cancer stem–like cells [50]. Hsp27 phosphorylation is enhanced in ALDH1+ BC stem cells (BCSCs), and it is essential for BCSC activity in multiple cancer types [51]. Furthermore, phosphorylated Hsp27 regulates cancer progression by epithelial–mesenchymal transition and NF-B activity, thus participating in the maintenance of stem cells [51]. In the tissues and cells of estrogen receptor-positive BC, Hsp27 is highly expressed in parallel with increasing histological grade and increasing SUMOylation of HSPB8 [52]. The depletion of Hsp27 reduces tumor formation rate and promotes cell apoptosis in breast cancer MCF-7 cells [53]. Hsp27 is highly upregulated in angiogenic BC cells, which suggests that the chaperone plays a key role in the balance between tumor dormancy and the expansive tumor growth associated with the onset of angiogenesis. Downregulation of Hsp27 in angiogenic BC cells induces long-term dormancy in vivo [54]. In addition, the experimental evidence demonstrates that Hsp27 and its phosphorylation are critical in the epidermal growth factor (EGF)-induced vasculogenic activity of BC stem/progenitor cells [55]. However, Hsp27 is expressed not only in tumor cells, but also in antigen-presenting cells, thus contributing to the immune escape mechanism in BC and favoring the differentiation of dendritic cells biased to inducing immune tolerance rather than response [56].

Taken together, the available data indicate that Hsp27 participates in mechanisms that promote the proliferation and metastasization of BC cells. Thus, blocking Hsp27 appears as a promising anti-BC chaperonotherapy strategy, a possibility deserving investigation.

### 3.2. Hsp60

The chaperonins chaperonin-containing TCP-1 (CCT), also called TCP-1 Ring Complex (TRiC), and Heat shock protein 60 (Hsp60) are molecular chaperones ranging in size from 55 kDa to 64 kDa. Like all molecular chaperones, they are typically cytoprotective, but if abnormal in quantity and/or quality they may be pathogenic, causing diseases named chaperonopathies, hereditary or acquired, including cancer [9]. The CCT family includes 14 genes, of which 9, namely CCT1, 2, 3, 4, 5, 6 (with two paralogues, 6A and 6B), 7, and 8 encode the subunits that form the functional hexadecamer involved in the chaperoning of many proteins [57]. The role of CCT in carcinogenesis, including BC, has been discussed elsewhere [58].

Although Hsp60 is considered a mitochondrial chaperone in eukaryotes, it has become clear that it also occurs in the cytosol, the cell surface, extracellular space, and plasma [58,59]. Thus, in addition to its role in polypeptide folding in association with Hsp10, other functions and interacting molecules have been identified for Hsp60. Some of these newly identified functions are associated with carcinogenesis, in particular cancer-cell survival and proliferation [58] (Figure 2). Hsp60 has been found elevated by two-dimensional gel electrophoresis in BC (ductal carcinomas), compared to normal breast tissues [60]. Furthermore, the levels of Hsp60 in BC tissues were shown to be elevated by immunohistochemistry and associated with increased tumor aggressiveness and poor prognosis [61].

In addition to assessing the levels of the Hsp60 protein, mapping its tissue and cellular distribution can provide distinctive patterns to identify tumors and degrees of malignancy. Hsp60 occurs not only inside mitochondria, but also on the surface of cancer cells, a location considered a danger signal for the IS, triggering the activation and maturation of dendritic cells and the generation of an antitumor T-cell response [62,63]. Thus, quantifying and mapping Hsp60 in BC tissue has potential as a diagnostic marker, prognostic indicator, and target for negative chaperonotherapy [61].

Detection of autoantibodies against Hsp60 in cancer patients constitutes an additional parameter to consider for early diagnosis [64]. Hsp60 mRNA levels were found to be significantly higher in primary BC compared to healthy breast tissues [64]. Thus, Hsp60 overexpression during the first steps of breast carcinogenesis can be considered to be, as a working hypothesis, functionally correlated to tumor growth and/or progression. However, it must be borne in mind that the potential of Hsp60 level evaluations for diagnosing and assessing the prognosis of breast tumors varies depending on the method used and on tumor type [64].

### 3.3. Hsp70

The Hsp70/DnaK family of proteins is composed of 17 members that reside in the various cell compartments [65]. For example, Hsc70 (HSPA8) works in the cytosol and nucleus, Grp78 or Bip (HSPA5) in the endoplasmic reticulum, and Hsp70-9B or mortalin (HSPA9) in mitochondria. HSPA8 (constitutive) and HSPA1A (Hsp70-1 inducible) are the two main isoforms of Hsp70 residing in the cytoplasm and nucleus. These Hsp70 chaperones carry out various tasks, interacting with other members of the CS to build chaperoning teams (e.g., the Hsp70/DnaJ chaperoning machine) and integrate functional networks (e.g., Hsp70-DnaJ machine/prefoldin/CCT), and they also interact with the collaborators of the CS, namely the UPS and the CMA machinery [1]. Their specific functions are: (i) binding and releasing misfolded proteins and interacting with prefoldin and CCT to assist in protein folding; (ii) transporting proteins across membranes to deliver them to organelles; (iii) ushering defective or damaged proteins toward the UPS for degradation; and (iv) bringing proteins into the endosome/lysosome for CMA [1,66]. For example, Hsp70 forms a team with Hsp40/DnaJ and NEF (nucleotide exchange factor), and this team then forms a network with Hsp90 [66]. The molecular structure of all Hsp70s consists of the N-terminal nucleotide-binding domain (NBD), which has the ATP-binding site and the substrate-binding domain (SBD) with a peptide-binding pocket and a bendable lid for clasping the substrate (peptide). NBD and SBD are connected by a flexible linker [66]. ATP binding to the NBD induces the open conformation of the SBD, which confers a low affinity for the binding of client proteins. The hydrolysis of ATP promotes the closed conformation and entrapment of client proteins with high affinity by the SBD [67,68].

Elevated Hsp70 levels have been found to correlate with lymph-node metastases and decreased survival in BC [69]. HSPA1A can be exposed on the surface of malignant cells and be secreted in EVs [70]. Intracellular HSPA1A protected tumor cells from immune attack in both in vitro and in vivo experiments [71] (Figure 2). In MDA-MB-231 cells (triple-negative human breast cancer cells (TNBC) ER-/PR-/HER2-), the inhibition of extracellular Hsp70 (isoform not identified in the original publication) suppresses tumor invasion of surrounding normal tissue through the Hsp90-dependent activation of matrix metalloproteinase-2 (MMP-2) [72]. Hsp70-2, a member of the cancer testis family of antigens (CTA), has been found to be associated with various malignancies, and, in particular, high expression levels have been observed in BC irrespective of stage, grade, HSPA1A, and histotype. Hsp70-2 expression was observed in breast cancer cell lines with different receptor statuses: BT-474 and SK-BR-3 (HER2-positive), MCF-7 (ER-positive), and MDA-MB-231, but not in human normal mammary epithelial cells HNMEC [73]. Hsp70-2 would be involved in the pathogenesis of BC by promoting viability, motility, migration, and invasion. In MDA-MB-231 and MCF7 cell lines, the depletion of Hsp70-2 by gene silencing resulted in the upregulation of caspases, pro-apoptotic molecules, and cyclin-dependent kinase (CDK) inhibitors and epithelial markers, while anti-apoptotic molecules, cyclins, CDKs, and mesenchymal markers were downregulated [73]. Hsp70-2 knockdown produced apoptosis, loss of colony-forming ability, and inhibition of growth of MDA-MB-231 xenografts [73]. Additionally, the knockdown of Hsp70-2 upregulated the p53/p21 pathway [74]. Similarly, to Hsp70-2, the knockdown or inactivation of Grp78 (HSPA5) suppressed migration and invasion in diverse BC cell lines [75,76]. Grp78 overexpression in MDA-MB-231 cells caused increased resistance to the chemotherapeutic drug cisplatin [75]. It was found that Grp78 is apparently required not only for tumor initiation, but also for tumor progression by regulating tumor angiogenesis [77]. Grp78 is a potential target for selective attack with anti-Grp78 antibodies. For example, it was reported that the binding of anti-Grp78 antibodies to Grp78 on the surface of BC MCF-7 cells suppressed their proliferation and migration mediated by STAT3 phosphorylation [78].

Mortalin (Grp75) is known to contribute to the development and pathogenesis of BC. It can promote metastasization and angiogenesis in DC cell lines, e.g., MCF-7, and SK-BR-3 lines [79,80]. The knockdown of Mortalin in MCF-7 and SK-BR-3 cells leads to overexpression of epithelial markers, such as ZO-1 and E-cadherin; however, mesenchymal markers showed low expression [79,80]. Mortalin promoted the endothelial-to-mesenchymal transition (EMT) in BC via the Wnt/β-catenin signaling pathway [79]. Mortalin is considered a molecular target for inducing complement-dependent cytotoxicity against BC cells [81,82]. Mortalin-mimetic peptides can be combined, as adjuvants, with complement-activating antibodies for treating BC [82]. In summary, members of the Hsp70/DnaK family are involved in the carcinogenic mechanisms in BC (Figure 2), and they can therefore be considered potential targets for treating BC via negative chaperonotherapy, namely the inhibition/blocking of the pathogenic chaperones.

### 3.4. Hsp90

The Hsp90 family is an important component of the CS and, in humans, is composed of four members that display canonical functions pertaining to the maintenance of protein homeostasis and have been intensively investigated in relation to carcinogenesis [5,83,84,85]. The four members reside in the following cell compartments: (i) Hsp90 alpha and Hsp90 beta are found in the cytosol; (ii) Grp94 (94 kDa glucose-regulated protein) resides in the endoplasmic reticulum; and (iii) TRAP-1 (tumor necrosis receptor-associated protein 1) is localized in the mitochondria [5,83,84]. The basic molecular structure of Hsp90 comprises three conserved domains, i.e., a carboxy-terminal domain (CTD), amino-terminal domain (NTD), and medium domain (MD). Each of these domains has specific functions, e.g., NTD is involved in ATP binding, CTD is the dimerization domain involved in binding to co-chaperones and client proteins, and MD modulates ATPase activity and binds co-chaperones and client proteins. Hsp90 also has the Charged Linker Region (CLR), which increases the protein’s flexibility and dynamism [85]. Hsp90 does not act alone, but rather forms teams with co-chaperones and networks with other chaperones, Hsp70, for instance, as mentioned earlier. Examples of Hsp90 co-chaperones are the Hsp70/Hsp90 organizing protein (HOP), cell division cycle 37 (Cdc37), FK506-binding protein 5 (FKBP5), carboxyl terminus of HSC70-interacting protein (CHIP), cyclophilin 40 (CYP40), and p23. The assembled team undergoes ATP hydrolysis cycles to exercise its chaperoning functions [86]. Cdc37 inhibits ATP hydrolysis by binding between the N-terminal domains of Hsp90 dimers, thus prolonging the association of the two Hsp90 molecules, boosting the chaperoning cycle [87,88]. Cdc37 binding to Hsp90alpha and beta inhibits their ATPase activity, which prolongs the interaction between chaperone and client. Furthermore, during the interaction with Hsp90, Cdc37 binds to the client kinase and to Hsp90 itself, with both interactions required for chaperone function [89]. The oncogenic receptor tyrosine kinase ERBB2 requires persistent association with Hsp90-Cdc37 complexes for stability and activity [89]. Hsp90 (isoform not specified in the original publication) can undergo post-translational modifications such as acetylation, phosphorylation, and S-nitrosylation, all of which have important repercussions on chaperoning activity [90].

In BC, Hsp90 plays an important role in the stabilization of several proteins that are important in cancer growth and survival (Figure 2). Clients of Hsp90 are ER, PR, components of HER2 signaling (such as HER2, AKT, RAF, c-SRC, and HIF-1α), and EGFR [91] (Figure 2). High levels of Hsp90alpha, Hsp90beta, and HSF1 correlated with poor prognosis in TNBC and HER2−/ER+ [92]. HSF-1, an important transcription factor in the regulation of Hsp90 and Hsp70 expression levels, is over-expressed in BC, where it promotes tumor progression by driving gene transcription in ER+ BC [93]. High levels of HSF-1 and its mRNA are associated with poor prognosis in BC [93]. Hsp90 (isoform not specified in the original publication) levels were assessed along the entire continuum of breast ductal lesions including ductal hyperplasia without atypia (DHWithoutA), atypical ductal hyperplasia (ADH), invasive ductal carcinoma (IDC), and ductal carcinoma in situ (DCIS) [94,95]. It was found that Hsp90 levels were higher in IDC, but DHWithoutA, ADH, and DCIS did not show levels of Hsp90 higher than in the adjacent normal tissue. High levels of Hsp90 (isoform not specified in the original publication) were observed in grade 2/3 IDCs and tumors of larger size. Increased levels of Hsp90 (isoform not specified in the original publication) in IDC correlated with higher grade, larger tumor size, and higher ER expression and c-erbB-2 positivity, while TNBC-IDC showed reduced levels of Hsp90 [95]. Hsp90alpha and beta in BC participate in the stabilization of the nuclear transcription factor E2F1/2 [96]. The inhibition of Hsp90alpha and beta with 17AAG (17-N-allylamino-17-demethoxygeldanamycin) caused destabilization of E2F1/2 transcription factors in MCF-7 cells [96]. The levels of Hsp90alpha and beta were assessed in MDA-MB-231 via CRISPR/Cas9, and a protective effect of Hsp90alpha in preventing cell death caused by hypoxia was observed; Hsp90beta, on the other hand, did not inhibit cell death caused by hypoxia or correct the lack of Hsp90alpha in Hsp90alpha-knockout clones [97]. In contrast, no Hsp90beta-knockout clones were obtained, suggesting that it is indispensable to cancer cell survival. High levels of Hsp90 (isoform not specified in the original publication) were found in ductal carcinomas, and they correlated with grade 2/3 IDC, larger tumor size, higher ER expression, and c-erbB-2 positivity [95]. In contrast, a decrease in Hsp90 (isoform not specified in the original publication) levels was observed in lobular carcinomas and lobular neoplasia [98]. This suggests that Hsp90 cannot be considered a universal marker for BC. HER2 overexpression promotes the activation of the HSF1-Hsp90 axis and the stabilization of Hsp90 client proteins, such as MIF, AKT, and mTOR, leading to increased tumor growth [99] (Figure 2). In contrast, HER2/ErbB2 suppression inhibited HSF-1, which led to blocking of Hsp90alpha chaperoning activity and tumor growth. HER2 signals via the PI3K-AKT-mTOR axis to activate HSF1 and the decrease in chaperone activity confirm the link between HER2 and HSF-1 [99]. Grp94 modulated intracellular trafficking of HER2 and promoted its stabilization at the plasma-cell membrane in tumor cells, so its overexpression has been implicated in HER2+ BC pathogenesis [100]. Furthermore, Grp94 participated in the regulation of ER-α36 (a variant of ER-α) expression levels, and its overexpression in BC was implicated in mechanisms of resistance to chemotherapeutic drugs [101]. High expression levels of Hsp90AA1alpha were associated with the EMT process [102]. It was observed that, after inhibiting Hsp90 (isoform not specified in the original publication) with penisuloxazin A, the epithelial-cell molecular marker E-cadherin increased, whereas the mesenchymal-cell molecular marker N-cadherin and vimentin and MMP9 proteins decreased significantly [103]. The examples mentioned in the preceding paragraphs indicate that Hsp90 plays an important role in the stabilization of several proteins that are involved in tumor growth, survival, and progression mechanisms in BC. Consequently, developing anti-Hsp90 agents is a promising anti-BC negative chaperonotherapy strategy, as discussed in the following section.

## 4. Hsp27, Hsp60, Hsp70, and Hsp90 in BC Therapeutics: Negative Chaperonotherapy

BC treatment is multidisciplinary, including local (e.g., surgery and radiotherapy) and systemic (e.g., chemotherapy and hormone therapy) approaches, depending on the cancer subtype and stage [104]. Chemotherapy is the main therapeutic approach, but it has negative aspects, including the development of drug resistance. Consequently, alternatives must be found to supplant or complement chemotherapy, for instance, those that use EVs and those based on chaperone inhibitors. The latter fall within the discipline of chaperonotherapy. Chaperonotherapy means the use of molecular chaperones and other components of the CS as therapeutic means or targets for therapeutic drugs to treat chaperonopathies, namely diseases in which chaperones and other components of the CS play an etiopathogenic role, such as in BC [105]. There are essentially three types of chaperonopathies: by defect, by excess, and by mistake [106]; consequently, there are corresponding types of chaperonotherapies: positive and negative. The former pertains to chaperonopathies by defect, in which the defective chaperone must be replaced or boosted. Instead chaperonopathies by excess (e.g., gain of function; or pathogenic increase in concentration, possibly associated with post-translational modifications and mislocalization) or by mistake must be eliminated or blocked. The latter applies to BC treatment as discussed later.

### 4.1. EVs

Exosomes are a class of EVs consisting of nano-sized membrane vesicles released by a variety of cell types that are thought to play important roles in intercellular communications [107]. Despite EVs not being considered to be targets for chaperonotherapy, they could be used as important diagnostic tool in cancer. For example, EVs transport components of the CS, e.g., molecular chaperons such as Hsp60, so that they can reach the various tissues in which they will ultimately function, thereby ensuring the coordinated performance of the CS [58]. Tumor cells release exosomes with cargo that includes potentially useful biomarkers and molecules with the ability to interact with recipient cells and modify their phenotype toward favoring metastasization [108,109].

In BC, exosomes are emerging as local and systemic mediators of oncogenic information through the horizontal transfer of bioactive molecules, such as enzymes, mRNA, and proteins, including Hsps [110]. Hsps can be released by tumor cells via exosomes, thereby modulating the local tumor microenvironment and preparing metastasization niches far away from the tumor [111]. Various Hsps have been found in the extracellular space or on the cell membranes [112]. Exosomes carrying Hsps can be isolated from diverse human body fluids, such as plasma, urine, and breast milk, making them convenient targets for early diagnosis and disease monitoring with minimally invasive procedures [112,113]. Elevated levels of Hsp27, Hsp60, Hsp70, and Hsp90 are associated with a poor prognosis in BC [111]. Hsp60 and Hsp90 are present on the surface of tumor cells and are secreted by these cells via exosomes, which makes these two chaperones promising, easily accessible biomarkers [114,115]. Likewise, the chaperones Hsp10 and Hsp27 have also been implicated in multifunctional chaperone networks in invasive BC [116].

A clinical prospective pilot study showed that the concentration of an unspecified isoform of Hsp70-positive exosomes in the plasma of BC patients was significantly higher compared to the plasma of healthy volunteers [117]. Assessing plasma-derived Hsp70-positive exosomes was found to be a more effective tool for differentiating patients with and without metastasis than the evaluation of circulating tumor cells (CTCs).

Another class of potential biomarkers present in circulation are small noncoding microRNAs (miRNAs), which play roles in the regulation of gene expression [118]. miRNAs secreted from exosomes improved the migration capacity of several BC cell lines [119,120]. miR-105, miR-122, and miR-106b are characteristically expressed and secreted by metastatic BC cells, promoting invasion and metastasization [110,121]. Furthermore, miRNAs may deregulate Hsps, serving as potential predictive and prognostic biomarkers [122]. Hsps are possible targets of miRNAs, causing their repression or deregulation, actively promoting apoptosis, cancer stemness, metastasization, and chemoresistance [122]. For example, the inhibition of miR-29a induces apoptosis by increasing the expression of Hsp60 and decreasing Hsp90, Hsp70, Hsp27, and Hsp40 levels in BC cells [123,124]. Investigation of miRNAs regulatory roles in the expression of Hsps with a role in carcinogenesis or with potential for chaperonotherapy warrants pursuit, because it may provide key information useful for understanding the participation of the CS in carcinogenesis and, thereby, facilitating the development of diagnostic tools and novel strategies for chaperonotherapy.

### 4.2. Hsp Inhibitors

When a chaperone’s functions are pro-carcinogenic, the condition can be classified as a chaperonopathy by mistake or collaborationism [106]. The chaperone has turned against the organism it must protect and collaborates with the enemy, so to speak. It is clear that, in such cases, a therapeutic strategy must be conceived to block the chaperone’s functions. This would be an example of negative chaperonotherapy, which contrasts with positive chaperonotherapy that consists of replacing or functionally boosting a defective chaperone. Hsp inhibitors have been used in negative chaperonotherapy with promising results [125]. Various inhibitors are currently being investigated at different stages of preclinical and clinical studies as novel anticancer agents. Hsp27 is overexpressed in BC, which makes this chaperone an attractive target for negative chaperonotherapy. The Hsp27 inhibitor antisense oligonucleotide OGX-427 is a specific inhibitor of Hsp27 that can be safely administered in patients and is currently in phase II clinical trials in BC and other cancers [126]. OGX-427 decreases the expression of Hsp27 [126] (Table 1).

Hsp27 participates in the maintenance of BC stem cells (BCSCs), and it has been noted that quercetin downregulates Hsp27 expression with the consequent suppression of BCSCs cells. It follows that Quercetin can potentially be employed in the chemoprevention of BC [51,127] (Table 1). This study shows that some compounds may not only act as complementary treatments to radio-chemotherapy but may also have potential as anticancer drugs on their own.

The Hsp60-Hsp10 chaperone complex plays an essential role in the maintenance of protein homeostasis in mitochondria in healthy organisms and in cancer cells. In the latter, Hsp60 is essential for maintaining the functions of the mitochondria at an accelerated rate, higher than in normal cells [140]. Therefore, this chaperonin appears as an attractive target to stop or slow down the metabolism and growth of the tumor cells. Hsp60 inhibitors can be divided into two major groups: type I, which block the binding and hydrolysis of ATP, and type II, which includes compounds that covalently bind certain cysteine residues in Hsp60 [141]. Epolactaene and its synthetic derivative ETB bind Hsp60 Cys442 and inhibit the chaperone’s activity [128] (Table 1). O-carboranylphenoxyacetanilide, an inhibitor of the hypoxia-inducible factor 1 alpha (HIF-1α) and its derivatives, binds Hsp60 and can inhibit the folding activity of Hsp60-Hsp10 and the ATPase activity of Hsp60 [142]. Another class of synthetic compounds that are used to inhibit Hsp60 are the porphyrin gold (III) complexes [143]. The gold (III) complex [Au (TPP)Cl] can interact with Hsp60. It is hypothesized that the gold (III) ion interacts electrophilically with Hsp60 and that porphyrin binds to Hsp60 via hydrophobic interactions [143].

The critical role of Hsp70, as discussed earlier regarding protein regulation and BC progression, suggest that this protein can also be used as a drug target with anti-BC effects. VER-155008 is an adenosine-derived compound that works by inhibiting the chaperoning activity of Hsp70 (unfortunately the isoform is not specified in the original work) by binding the ATPase domain [129]. The ability of VER-155008 to induce apoptosis in MCF-7 breast cancer cells has been shown in in vitro studies [129] (Table 1). Treatment with embelin reduced cancer-cell growth and metastatic potential by activating p53 in BC [144]. YK5 is a specific Hsp70 interactor with biological activity in part through interfering with the formation of active Hsp70/Hsp90/client oncogenic protein complexes (e.g., HER2 and Raf-1) [130]. YK5 is a small-molecule inhibitor rationally designed to interact with an allosteric site located in the ATP-binding domain of Hsp70 and serves as a chemical tool for studying the cellular mechanisms associated with Hsp70 [131] (Table 1). Another highly selective and novel allosteric inhibitor of Hsp70 is HS-72 (purine-like molecule), which reduces ATP-binding affinity, inhibits tumor growth, and increases survival in a BC animal model [132] (Table 1). DMT3132 is an analog of MAL3-101 (a family of pyrimidinones) that showed antiproliferative activity in BC cells [133] (Table 1). A flavonoid derived from licorice root, isoliquiritigenin, directly targets Grp78 and suppresses cell-colony formation by BC stem cells [134] (Table 1).

Several compounds that inhibit Hsp90 with promising anticancer properties in preclinical studies have been described [145]. Geldanamycin is an antibiotic that can bind Hsp90 and inhibit its chaperoning activities, but its toxicity has led to the development of analogues, for example 17-AAG (or tanespimycin). Tanespimycin has proven safe and effective in combination with TZMB for the treatment of refractory HER2-positive BC with a response rate of 22% and a clinical benefit rate of 59% [135] (Table 1). Other Hsp90 inhibitors have been developed, including more water-soluble analogues, such as 17-DMAG (alvespimycin), IPI-504 (retaspimycin), and several new synthetic small molecules with different properties. The treatment of HER2 (+) BC cells with acquired resistance to lapatinib (a tyrosine kinase inhibitor), using a combination of lapatinib and 17-DMAG suppressed the growth of BC cells in vivo and in vitro [136] (Table 1). Clinical data have demonstrated effective HER2 targeting in patients with HER2-positive BC refractory to TZMB when treated with IPI-504 in combination with TZMB, with modest anti-tumor effects [137] (Table 1). The Hsp90 inhibitors described above are part of the first-generation inhibitors. The second generation Hsp90 inhibitors are less toxic, e.g., Ganetespib (5-(2,4-dihydroxy-5-(1-methylethyl)phenyl)-4-(1-methyl-1H-indol-5-yl)-2,4-dihydro-(1,2,4)triazol-3-one), that has produced promising antitumor effects with acceptable safety profiles. Ganetespib binds to the ATP pocket in the *N*-terminus of Hsp90alpha and Hsp90beta [138]. Ganetespib showed antitumor activity on the BC subtypes HER2-normal and TNBC, with a favorable safety profile, including lack of hepatotoxicity and ocular toxicity [138] (Table 1). A novel approach to the treatment of metastatic BC is the combination of Hsp90 inhibitors and taxanes. The combination of Ganetespib with paclitaxel and TZMB is well tolerated and safe in a heavily pretreated population of HER2-positive metastatic breast cancer [139] (Table 1). In addition to Ganetespib, there are other second-generation Hsp90 inhibitors that are in preclinical or clinical testing in patients with BC, such as NVP-AUY922 (resorcinol derivative) [146], PU-H71 (purine derivative) [147], and other inhibitors of the benzamide class, including XL888.

## 5. Hsps in BC Immunotherapy

In the anti-cancer immune response, a release of cancer-cell antigens is the first step [148]. Chemotherapy suppresses the immune response, but there are anti-cancer agents that can induce cell death through immune mechanisms [149]. Mammary tumor cells escape immune surveillance by several mechanisms, including tumor-specific antigens (TSAs) as HER2 or releasing tumor-factors that contribute to tumor progression [150]. In the interstitial fluid of BC there are often elevated levels of Hsps that are considered key components of the tumor microenvironment [72,97], and, in this location, the Hsps could be subverting anti-tumor immunity [147,151]. For instance, Hsp27 was present at high levels in the interstitial fluid of human BCs, where it was involved in the differentiation of human monocytes to macrophages and in the induction of tolerogenic and pro-angiogenic tumor-associated macrophages [152]. Elevated extracellular Hsp70 levels could propagate chronic inflammation, which could in turn suppress the antitumor immune responses [72]. Hsps have been reported to elicit specific humoral immunity in BC patients when released into the extracellular environment [80,153]. Antibodies specific for Hsps have been found in the blood of women with BC with improved survival [154]. The presence of autoantibodies against Hsp60 was reported in 31 and 32.6% of early BC sera and DCIS, respectively, compared to 4.3% in healthy controls, suggesting that chaperonin plays a role in early carcinogenesis, and, therefore, Hsp60 may be considered of interest in the development of immunotherapeutic approaches [155]. The inducible Hsp70 expressed on the surface of BC cells can be targeted by specific anti-Hsp70 antibodies [156]. Autoantibodies against Hsp90 were identified in sera of BC, although the Hsp90 isoforms studied were not specified in the original work [157,158]. Extracellular Hsps can bind and present tumor-associated antigens to professional antigen-presenting cells (APCs) interacting with MHC class I and class II molecules, thus leading to the activation of anti-tumor T cells [148,159]. Since BC contains high levels of Hsps that accompany the expanding levels of oncoproteins, they are a convenient source for collecting Hsp-antigen complexes [160]. The purified Hsp-peptide complexes represent the antigenic fingerprint of the cell from which they are isolated. This property is desirable for the preparation of cancer vaccines, although they require an important therapeutic intervention, unlike vaccines for other diseases. Hsps are seen as important anticancer-vaccine adjuvants, and they are used with different delivery systems, e.g., Hsps/antibodies, peptide/protein-Hsp complexes, tumor antigen/HSP gene fusion, and viral peptides/Hsp complexes [161]. Several clinical studies have been carried out or are ongoing, immunizing cancer patients with autologous tumor-derived Hsp-peptide complexes (HSPPCs) [162]. Clinical trials have been carried out with Hsp-based vaccines and the results suggest that the approach is safe and potentially effective; however, thus far, no clinical trials have been carried out with BC [162]. In one approach, Hsp70 molecules were extracted from tumors by affinity chromatography in association with a repertoire of tumor antigens and used as autologous vaccines [163]. This approach has the advantage of a personalized, autologous vaccine, carrying a range of antigenic epitopes, also with adjuvant activity, helping innate immune stimulation by the Hsp itself [163]. The p391 and p393 peptide sequences from inducible Hsp70 have a high affinity for the major histocompatibility complex HLA-A*0201 and are thus able to activate cytolytic T-lymphocytes. However, these Hsp70 epitopes may not be the targets of an immune response in many HLA-A*0201+ BC [164]. Another approach involved testing Hsp70s associated with tumor-cell-derived proteins or peptides in an aggressive spontaneous metastatic tumor model of BC [165]. Hsp70 molecules were extracted from dendritic cells (DCs) fused with patient-derived BC cells, known as Hsp70.PC-F or Hsp70.peptide complexes from fusion cells (HSP70.PC-F), to produce a vaccine [166]. The resulting vaccine was a potent T cell stimulator with the ability to produce high interferon-γ and kill murine tumors in vivo and human breast cancer cell lines in vitro [166] (Table 2).

This procedure, using pooled DC and BC cell lines to prepare tumor DC fusions has potential as a universal BC vaccine, because it is enriched in MUC1 and HER2/neu antigens [166]. The Hsp70.PC-F vaccine also contains coprecipitating Hsp90, which is essential for tumor regression [166]. The Hsp70-peptide complexes isolated from DCs fused to 4T1 murine BC cells were encapsulated in nanoliposomes to improve their bioavailability and antitumor immunity, resulting in an increase in the activation of T lymphocytes in BC, compared to the control variant (without liposomes) [167] (Table 2). In another study, a peptide aptamer that binds to the extracellular domain of the Hsp70 membrane and targets Hsp70-expressing exosomes in BC patients was used; this is because the Hsp70 molecules exposed on the exosome membrane are able to interact with the Toll-like receptor 2 on the surface of myeloid-derived suppressive cells, thereby activating the latter and protecting tumor cells from immune recognition [168]. Vaccination with autologous tumor-derived Hsp70-peptide complexes is in clinical phases I and II in high-risk BC patients [169].

The studies mentioned in this section highlight the necessity of developing novel anticancer therapies by combining different immunotherapeutic methods. There is accumulating evidence that effective cancer immunotherapy must activate the IS in multiple manners to obtain a solid and long-lasting immune response. Lastly, it should be mentioned that Hsp molecules have also been utilized in DNA-based vaccines as antigens or adjuvants against cancer. For example, the NeuEDhsp70 DNA vaccine induced enhanced immune responses, which conferred anti-tumor immunity to 4T1.2-Neu BC, an aggressive, spontaneous, metastatic BC [165].

## 6. Conclusions and Perspectives

Despite intensive research over many years, BC remains a serious public health problem. New strategies must be developed to improve early diagnosis, treatment, and patient monitoring. The CS is emerging as a determinant factor in mammary carcinogenesis, and elucidating its participation in the various steps of tumor development, encompassing initiation, progression, dissemination, resistance to treatment, and recurrence after a period of remission is important. Therefore, the study of CS participation in BC is expected to provide a platform and a guide for developing novel diagnostic and prognostication tools and efficacious treatments. The chief components of the CS are the molecular chaperones; in this review, we discuss four of them: Hsp27, Hsp60, Hsp70, and Hsp90. Their role in BC carcinogenesis and potential as therapeutic targets or tools as reported in the literature are presented. Chaperones are typically cytoprotective, but they can also be pathogenic, for example, when they sustain the protein homeostasis of cancer cells, and can grow and spread with high efficiency. This is an example of chaperonopathy by mistake or collaborationism: the chaperone turns against the organism that it should protect. Consequently, negative chaperonotherapy is required: the pathogenic chaperone must be eliminated or blocked. For this purpose, many Hsp inhibitors have been and are currently being developed and tested. It is hoped that an efficacious inhibitor with minimal side effects will soon be found. Some Hsps have been tested as adjuvants with BC antigens in immunotherapy, and the results are promising. In summary, studying the CS in the pathogenesis of BC and targeting some of its components, such as Hsps, for use as anticancer drugs for negative chaperonotherapy are two very promising research lines that are expected to produce practical results in the near future. Similarly, the study of EVs as carriers of Hsp biomarkers useful in diagnosis and patient monitoring and the use EVs for anti-cancer drug delivery are also promising research avenues. Likewise, using Hsps as adjuvants together with BC antigens is another aspect of the role of the CS that merits exploration, as the results thus far are encouraging.

## Figures and Tables

**Figure 1 ijms-23-07792-f001:**
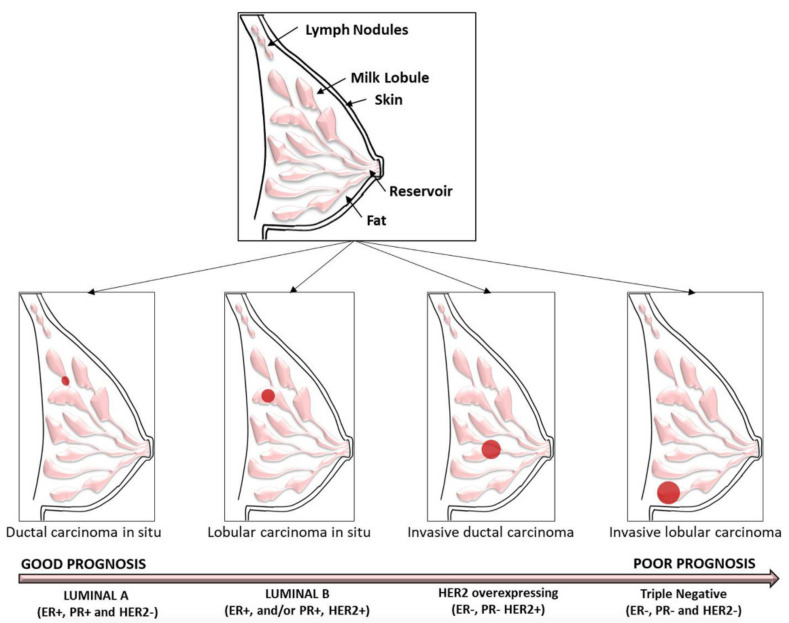
Drawings representing histological and molecular BC subtypes with corresponding prognostic implications (ER: Estrogen Receptor; PR: Progesterone Receptor; HER2: Human epidermal growth factor receptor 2). The red dots indicate the cancer localization.

**Figure 2 ijms-23-07792-f002:**
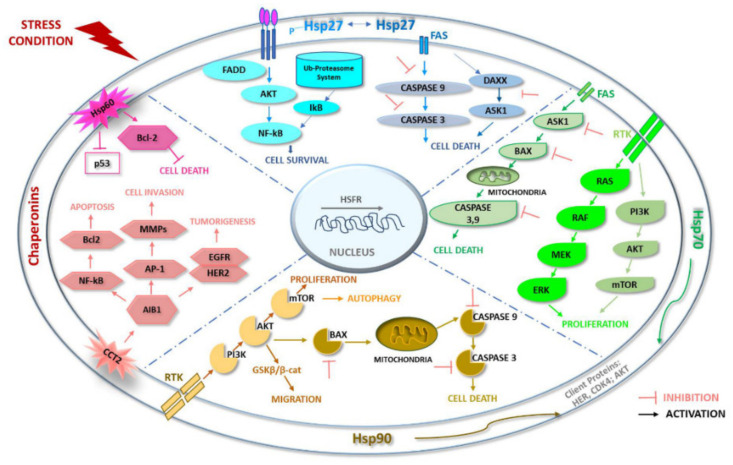
The CS in carcinogenesis. The chief components of the CS are the molecular chaperones, and the figure shows steps in carcinogenesis, e.g., in BC, in which they actively participate, favoring tumor growth, which makes them suitable targets for negative chaperonotherapy, namely eliminating/blocking/inhibiting the pathogenic chaperone. For instance, Hsp27 regulates apoptosis through interaction with Akt and the inhibition of caspases-3/9 activity, thereby leading to tumor cell survival. The T-complex protein 1 subunit beta (CCT2)-mediated AIB1 folding is involved in cancer development in pathways of cell growth, invasiveness, and survival. AIB1 promotes cancer development via hormone-dependent pathways, acting as a transcriptional coactivator for nuclear receptors in estrogen receptor (ER)-positive breast cancer. Moreover, several non-nuclear receptor transcription factors, e.g., HER-2, NF-kB and MMPs, are coactivated by AIB1. Hsp60 may function either as an anti-apoptotic factor promoting cancer cell survival, or as a pro-apoptotic factor that promotes cancer cell death. Hsp70 inhibits tumor-cell apoptosis by interfering with the formation of the apoptosome by preventing caspases-3/9 activation. Hsp70 also induces evasion from apoptosis, blocking PI3K/AKT signaling. Hsp90 is involved in cancer development in pathways of cell growth, invasiveness, and survival. The components of the PI3K/AKT/mTOR pathway are Hsp90 clients, and this chaperone thus plays a role in the regulation of autophagy with a tumor cell pro-survival function.

**Table 1 ijms-23-07792-t001:** Hsp inhibitors used alone or in combination with other therapeutic agents.

Inhibitor’s Target	Therapeutic Agent	Experimental Design and Trial Phase	Reference
Hsp27	OGX-427 OGX-427 + Quercetin	In vitro, Phase II In vitro	[126] [51,127]
Hsp60	Epolactaene	In vitro	[128]
Hsp70 (Hsp70 family member not specified)	VER-155008	In vitro	[129]
Hsp70 and Hsc70	YK5	In vitro	[130,131]
HSPA1A	HS-72	In vitro and in vivo	[132]
Hsp70 (Hsp70 family member not specified)	DMT3132	In vitro	[133]
Grp78	Flavonoid derivated	In vitro	[134]
Hsp90 (isoform not specified)	Tanespimycin + TZMB	In vitro, Phase II	[135]
Hsp90 alpha	Alvespimycin+ lapatinib	In vitro and in vivo	[136]
Hsp90 (isoform not specified)	IPI-504 + TZMB	In vitro, Phase II	[137]
Hsp90 alpha and Hsp90beta	Ganetespib	In vitro, Phase I	[138]
Hsp90 (isoform not specified)	Ganetespib + Paclitaxel	In vitro	[139]

**Table 2 ijms-23-07792-t002:** Hsps-based vaccines against breast cancer in clinical trials.

Hsp	Therapeutic	Trial Phase	Reference
Hsp70 (Hsp70 family member not specified)	Hsp70-peptide complexes	In vitro and in vivo, Phase I/II	[166]
Hsp70 (Hsp70 family member not specified)	Hsp70-peptide complex at 4T1	In vivo, Phase I/II	[167]
Hsp70 (Hsp70 family member not specified)	NeuEDhsp70 DNA	In vitro	[165]

## Data Availability

Not applicable.

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
