# Peer review of "The Chaperone System in Breast Cancer: Roles and Therapeutic Prospects of the Molecular Chaperones Hsp27, Hsp60, Hsp70, and Hsp90"

_ijms, 2022, doi:10.3390/ijms23147792_

Round 1
Reviewer 1 Report
Please see the attached file for the review report. (Comments are also given below.)
The review manuscript “The chaperone system in breast cancer: Roles and therapeutic prospects of the molecular chaperones Hsp27, Hsp60, Hsp70, and Hsp90 “ is well written and enjoyable to read.
Although the subject is interesting and comprehensively described some concerns need to be addressed before the manuscript is ready for publication.
Comments:
1. Not all abbreviations have been defined in parentheses the first time they appear in the text or in figure captions (e.g. “TNBC”, “HR”, “NOS”, “CCT2”).
2. I suggest to avoid using an abbreviation if it occurs only one time in the text (e.g. “IS”, “TZMB”, “TAM”, “CTL”).
3. Page 4, Figure 1, bottom: The description of the BC subtypes is not consistent with that given in lines 132-135.
4. Page 6, the figure caption: The number of the figure is wrong. It should be “Figure 2” instead of “Figure 1”. I also recommend a larger font size for words in the top panel of the figure.
5. Please see comment No. 4. Figure 2: Descriptions placed on dark blue and dark green shapes are not readable.
6. Please see comment No. 4. Figure 2, top: It should be “Ub-Proteasome System” or “Ubiquitin-Proteasome System” instead of “Ub. Proteosoma system”.
7. Please see comment No. 4. Figure 2 caption, line 232: I suggest “PI3K/AKT signaling” instead of “PI3K/Akt signaling”.
8. Table 1: It should be “Ganetespib” instead of “Gametes”. Please see ref. [138].
9. Table 1: I suggest the same way of writing the names of the therapeutic agents when the agents are administered in combination.
10. Table 2 title: It should be “Hsps-based” instead of “Hsps -based”.
11. Page 2, line 100: It should be “(Hsps)” instead of “(Hsp)”.
12. Page 2, line 104: I suggest “one representative of the sHsps” instead of “one representative of the sHsp”.
13. Page 2, line 117: I suggest “block/inhibit” instead of “block-inhibit”.
14. Page 2, line 131: It should be “human epidermal growth factor 2 receptor (HER2)“ instead of “factor 2 receptor human epidermal growth (HER2)”.
15. Page 4, line 145: It should be “HER2-positivity” instead of “ER2-positivity”.
16. Page 4, line 158: It should be “five subtypes, namely, comedo, solid, cribriform, papillary, and micropapillary.” instead of “four subtypes, namely, comedo, solid, cribriform, papillary, and micropapillary.”
17. Page 4, line 165: It should be “criteria, including” instead of “criteria. including”.
18. Page 6, line 251: I suggest “trastuzumab” instead of ”TZMB”. I recommend to avoid using an abbreviation if it occurs only one time in the text (please see also comment No. 2).
19. Page 8, line 322: It should be “Hsp70/DnaK” instead of “Hsp70/DnK”.
20. Page 8, line 424: I recommend “endoplasmic reticulum, and Hsp70-9B” instead of “endoplasmic reticulum (ER), and Hsp70-9B”. The Authors use “ER” as the abbreviation of “estrogen receptor” throughout the manuscript.
21. Page 8, line 347: It should be “TNBC, ER-/PR-/HER2-)” instead of “TNBC, ER-/PR-/Her2-)”. Please compare also with Figure 1.
22. Page 9, line 372: It should be “DC cell lines” instead of “dc cell lines”.
23. Page 9, line 374: It should be “E-cadherin” instead of “E-Cadherin”.
24. Page 9, line 381: I suggest “inhibition/blocking” instead of “inhibition-blocking”.
25. Page 9, line 391: Please see comment No.20 for “(ER)”.
26. Page 10, line 416: It should be “Figure 2” instead of “Figure2”.
27. Page 10, line 418: It should be “c-SRC” instead of “c -SRC”.
28. Page 10, lines 428-429: The statement “High levels of Hsp90 (isoform not specified in the original publication) were observed in 2/ 3 IDC and large size.” should be corrected.
29. Page 11, line 513: I suggest “circulating tumor cells (CTCs)” instead of “circulating tumor cells (CTC)”.
30. Page 13, lines 548-549: It should be “quercetin” instead of “Quercetin”.
31. Page 13, line 556: It should be “target to” instead of “targetto”.
32. Page 13, line 561: It should be “[128] (Table 1).” instead of “[128]. (Table 1).”
33. Page 14, line 611: It should be “derivative) [145]” instead of “derivative), [145]”
34. Page 15, line 663: It should be “dendritic cells (DCs)” instead of “dendritic cells (DC)”.
35. Page 15, line 666: Italics should be used for “in vivo” and “ in vitro” (see Table 2).
36. Page 16, lines 674-677: Please see comment No. 34 for “DC”.
37. Page 17, Abbreviations:
a. I suggest to add “CS”, “EVs”, “CMA”, and “UPS” to Abbreviations.
b. It should be “CTCs, circulating tumor cells” instead of “CTC, circulating tumor cells”. Please see comment No. 29.
c. “ICD, immunogenic cell death” should be removed. This term does not occur in the text.
d. The abbreviation “TZMB” occurs only one time in the text. Please see comments No. 2 and No. 18. I suggest to remove “TZMB” from Abbreviations.

Author Response
Comments and Suggestions for Authors
Reviewer #1 Comment #1 (R#1C#1): Not all abbreviations have been defined in parentheses the first time they appear in the text or in figure captions (e.g. “TNBC”, “HR”, “NOS”, “CCT2”).
Author Replay (AR): We thank the reviewer for this comment. We wrote the full name, as indicated in track changes.
R#1C#2: I suggest to avoid using an abbreviation if it occurs only one time in the text (e.g. “IS”, “TZMB”, “TAM”, “CTL”).
AR: We thank the reviewer for this comment. We have eliminated abbreviations that do not repeat.
R#1C#3: Page 4, Figure 1, bottom: The description of the BC subtypes is not consistent with that given in lines 132-135.
AR: We are sorry for the mistake! We corrected the image according to the description in lines 132-135.
R#1C#4: Page 6, the figure caption: The number of the figure is wrong. It should be “Figure 2” instead of “Figure 1”. I also recommend a larger font size for words in the top panel of the figure. AR: We thank the reviewer for this comment. We have corrected the numbering of the figure.
R#1C#5: Please see comment No. 4. Figure 2: Descriptions placed on dark blue and dark green shapes are not readable.
AR: We thank the reviewer for this comment. We replaced with lighter and more legible colours.
R#1C#6: Figure 2, top: It should be “Ub-Proteasome System” or “Ubiquitin-Proteasome System” instead of “Ub. Proteosoma system”.
AR: We thank the reviewer for this comment. We corrected it.
R#1C#7: Please see comment No. 4. Figure 2 caption, line 232: I suggest “PI3K/AKT signaling” instead of “PI3K/Akt signaling”.
AR: We thank the reviewer for this comment. We corrected it.
R#1C#8: Table 1: It should be “Ganetespib” instead of “Gametes”. Please see ref. [138].
AR: We thank the reviewer for this comment. We corrected it.
R#1C#9: Table 1: I suggest the same way of writing the names of the therapeutic agents when the agents are administered in combination.
AR: We thank the reviewer for this comment. We added the names of the therapeutic agents that work in combination where necessary
R#1C#10: Table 2 title: It should be “Hsps-based” instead of “Hsps -based”.
AR: We thank the reviewer for this comment. We corrected it.
R#1C#11: Page 2, line 100: It should be “(Hsps)” instead of “(Hsp)”.
AR: We thank the reviewer for this comment. We corrected it.
R#1C#12: Page 2, line 104: I suggest “one representative of the sHsps” instead of “one representative of the sHsp”.
AR: We thank the reviewer for this comment. We corrected it.
R#1C#13: Page 2, line 117: I suggest “block/inhibit” instead of “block-inhibit”.
AR: We thank the reviewer for this comment. We corrected it.
R#1C#14: Page 2, line 131: It should be “human epidermal growth factor 2 receptor (HER2)“ instead of “factor 2 receptor human epidermal growth (HER2)”.
AR: We thank the reviewer for this comment. We corrected it.
R#1C#15: Page 4, line 145: It should be “HER2-positivity” instead of “ER2-positivity”.
AR: We thank the reviewer for this comment. We corrected it.
R#1C#16: Page 4, line 158: It should be “five subtypes, namely, comedo, solid, cribriform, papillary, and micropapillary.” instead of “four subtypes, namely, comedo, solid, cribriform, papillary, and micropapillary.”
AR: We thank the reviewer for this comment. We corrected it.
R#1C#17: Page 4, line 165: It should be “criteria, including” instead of “criteria. including”.
AR: We thank the reviewer for this comment. We corrected it.
R#1C#18: Page 6, line 251: I suggest “trastuzumab” instead of ”TZMB”. I recommend to avoid using an abbreviation if it occurs only one time in the text (please see also comment No. 2).
AR: We thank the reviewer for this comment. We corrected it.
R#1C#19: Page 8, line 322: It should be “Hsp70/DnaK” instead of “Hsp70/DnK”.
AR: We thank the reviewer for this comment. We corrected it.
R#1C#20: Page 8, line 424: I recommend “endoplasmic reticulum, and Hsp70-9B” instead of “endoplasmic reticulum (ER), and Hsp70-9B”. The Authors use “ER” as the abbreviation of “estrogen receptor” throughout the manuscript.
AR: We thank the reviewer for this comment. We corrected it.
R#1C#21: Page 8, line 347: It should be “TNBC, ER-/PR-/HER2-)” instead of “TNBC, ER-/PR-/Her2-)”. Please compare also with Figure 1.
AR: We thank the reviewer for this comment. We corrected it.
R#1C#22: Page 9, line 372: It should be “DC cell lines” instead of “dc cell lines”.
AR: We thank the reviewer for this comment. We corrected it.
R#1C#23: Page 9, line 374: It should be “E-cadherin” instead of “E-Cadherin”.
AR: We thank the reviewer for this comment. We corrected it.
R#1C#24: Page 9, line 381: I suggest “inhibition/blocking” instead of “inhibition-blocking”.
AR: We thank the reviewer for this comment. We corrected it.
R#1C#25: Page 9, line 391: Please see comment No.20 for “(ER)”.
AR: We thank the reviewer for this comment. We corrected it.
R#1C#26: Page 10, line 416: It should be “Figure 2” instead of “Figure2”.
AR: We are sorry for the mistake! We corrected it.
R#1C#27: Page 10, line 418: It should be “c-SRC” instead of “c -SRC”.
AR: We thank the reviewer for this comment. We corrected it.
R#1C#28: Page 10, lines 428-429: The statement “High levels of Hsp90 (isoform not specified in the original publication) were observed in 2/ 3 IDC and large size.” should be corrected.
AR: We thank the reviewer for this comment. We corrected it.
R#1C#29: Page 11, line 513: I suggest “circulating tumor cells (CTCs)” instead of “circulating tumor cells (CTC)”.
AR: We thank the reviewer for this comment. We corrected it.
R#1C#30: Page 13, lines 548-549: It should be “quercetin” instead of “Quercetin”.
AR: We thank the reviewer for this comment. We corrected it.
R#1C#31: Page 13, line 556: It should be “target to” instead of “targetto”.
AR: We thank the reviewer for this comment. We corrected it.
R#1C#32: Page 13, line 561: It should be “[128] (Table 1).” instead of “[128]. (Table 1).”
AR: We thank the reviewer for this comment. We corrected it.
R#1C#33: Page 14, line 611: It should be “derivative) [145]” instead of “derivative), [145]”
AR: We thank the reviewer for this comment. We corrected it.
R#1C#34: Page 15, line 663: It should be “dendritic cells (DCs)” instead of “dendritic cells (DC)”.
AR: We thank the reviewer for this comment. We corrected it.
R#1C#35: Page 15, line 666: Italics should be used for “in vivo” and “ in vitro” (see Table 2).
AR: We thank the reviewer for this comment. We corrected it.
R#1C#36: Page 16, lines 674-677: Please see comment No. 34 for “DC”.
AR: We thank the reviewer for this comment. We corrected it.
R#1C#37: Page 17, Abbreviations:
AR: We thank the reviewer for this comment. We corrected it.

Reviewer 2 Report
In my opinion the authors made a very good work focusing their review on the chaperone system in breast cancer. I have only a few suggestions to authors:
- It would be better for the reader to find specific references after some statements. I listed the places I think reference(s) is/are lacking:
At line 55 after “De-52 spite the many reports on the levels of molecular chaperones, i.e., the chief components of the CS, in tumor tissue and a variety of samples from patients, and despite experiments to determine their role in the tumor’s biology….”
At line 59 after “We have undertaken research on the role of the CS in various tumors, BC in- cluded, starting from histological studies to quantify and map diverse chaperones in the tumor tissue, and continuing with detection of chaperones in EVs released by the tumors.”
At line 70 after “The CS also displays non-canonical functions unrelated to protein quality control but pertaining to other mechanisms. For instance, the CS interacts with the immune system (IS) in performing some of its non-canonical functions in cancer and inflammatory and autoimmune conditions”
At line 556 after “The Hsp60-Hsp10 chaperone complex plays an essential role in the maintenance of protein homeostasis in mitochondria in healthy organisms and in cancer cells. In the latter, Hsp60 is essential to maintain the functions of the mitochondria at an accelerated rate, higher than in normal cells.”
- In Lines 295-296 is it correct to use periferal blood? Serum and/plasma do not fit better?
- Authors may state in the beginning of section 4.1 that the extracellular vesicles are not a target for chaperonotherapy but it is included in this part of the review since it can be a very relevant diagnostic tool.
Author Response
Comments and Suggestions for Authors
R#2C#1: It would be better for the reader to find specific references after some statements. I listed the places I think reference(s) is/are lacking.
AR: We thank the reviewer for this comment. We entered the reference number 1, which is used to describe the sentence.
R#2C#2: At line 59 after “We have undertaken research on the role of the CS in various tumors, BC included, starting from histological studies to quantify and map diverse chaperones in the tumor tissue, and continuing with detection of chaperones in EVs released by the tumors.”
AR: We thank the reviewer for this comment. We added new references.
R#2C#3: At line 70 after “The CS also displays non-canonical functions unrelated to protein quality control but pertaining to other mechanisms. For instance, the CS interacts with the immune system (IS) in performing some of its non-canonical functions in cancer and inflammatory and autoimmune conditions”
AR: We thank the reviewer for this comment. We entered the reference number 1, which is used to describe the sentence.
R#2C#4: At line 556 after “The Hsp60-Hsp10 chaperone complex plays an essential role in the maintenance of protein homeostasis in mitochondria in healthy organisms and in cancer cells. In the latter, Hsp60 is essential to maintain the functions of the mitochondria at an accelerated rate, higher than in normal cells.”
AR: We thank the reviewer for this comment. We entered the new reference which is used to describe the sentence.
R#2C#5: In Lines 295-296 is it correct to use periferal blood? Serum and/plasma do not fit better?
AR: We thank the reviewer for this comment. We corrected it.
R#2C#6: Authors may state in the beginning of section 4.1 that the extracellular vesicles are not a target for chaperonotherapy but it is included in this part of the review since it can be a very relevant diagnostic tool.
AR: We thank the reviewer for this comment. We added the new sentence in section 4.1.
